# Signaling Pathways Regulated by UBR Box-Containing E3 Ligases

**DOI:** 10.3390/ijms22158323

**Published:** 2021-08-03

**Authors:** Jung Gi Kim, Ho-Chul Shin, Taewook Seo, Laxman Nawale, Goeun Han, Bo Yeon Kim, Seung Jun Kim, Hyunjoo Cha-Molstad

**Affiliations:** 1Anticancer Agent Research Center, Korea Research Institute of Bioscience and Biotechnology, Daejeon 28116, Korea; comices@kribb.re.kr (J.G.K.); seotaewook@kribb.re.kr (T.S.); lax01nawale@kribb.re.kr (L.N.); goeun531@kribb.re.kr (G.H.); 2Department of Biomolecular Science, KRIBB School, University of Science and Technology, Daejeon 34113, Korea; 3Disease Target Structure Research Center, Korea Research Institute of Bioscience and Biotechnology, Daejeon 34141, Korea; shinhc81@kribb.re.kr

**Keywords:** UBR Box E3 ligases, N-recognin, Arg/N-degron pathway, N-degron, G-protein signaling, apoptosis, mitochondrial quality control, inflammatory response, DNA damage

## Abstract

UBR box E3 ligases, also called N-recognins, are integral components of the N-degron pathway. Representative N-recognins include UBR1, UBR2, UBR4, and UBR5, and they bind destabilizing N-terminal residues, termed N-degrons. Understanding the molecular bases of their substrate recognition and the biological impact of the clearance of their substrates on cellular signaling pathways can provide valuable insights into the regulation of these pathways. This review provides an overview of the current knowledge of the binding mechanism of UBR box N-recognin/N-degron interactions and their roles in signaling pathways linked to G-protein-coupled receptors, apoptosis, mitochondrial quality control, inflammation, and DNA damage. The targeting of these UBR box N-recognins can provide potential therapies to treat diseases such as cancer and neurodegenerative diseases.

## 1. Introduction

A variety of mechanisms regulate cellular signaling pathways. One such mechanism is the control of protein degradation. Protein degradation serves as a protein homeostasis regulatory network that removes unnecessary proteins from the cellular environment when they are no longer needed, damaged, or misfolded. In eukaryotic cells, the ubiquitin–proteasome system (UPS) and the autophagic–lysosomal pathway are the two major protein degradation systems [1]. Of these, the UPS is responsible for the bulk of intracellular protein degradation (over 80%) and plays an essential regulatory role in critical cellular processes, including cell cycle progression, proliferation, differentiation, angiogenesis, and apoptosis [2,3,4,5,6]. The dysregulation of this pathway is associated with many conditions such as neurodegeneration, cancer, and aging [7,8,9,10,11,12,13].

The UPS utilizes ubiquitin, a 76-amino acid polypeptide, as a tag to mark substrates for degradation. This process is called protein ubiquitination and is mediated by the coordinated action of a cascade of enzymes, including ubiquitin-activating enzymes (E1s), ubiquitin-conjugating enzymes (E2s), and E3 ubiquitin ligases (E3s) [14,15,16,17]. Protein ubiquitination starts with an E1 enzyme which activates ubiquitin by adenylating its C-terminus. Once activated, ubiquitin is conjugated to an E2 enzyme. Finally, an E3 ubiquitin ligase transfers the ubiquitin from the E2 enzyme to the target protein (substrate). As a result, this process covalently links the C-terminal glycine of ubiquitin to a lysine residue of the target protein through the formation of an isopeptide bond. Thus, E3s are particularly critical players in the ubiquitination process because they determine substrate specificity. There are more than 600 human E3 ubiquitin ligases encoded by approximately 5% of the human genome [18,19,20].

One unique class of E3 ubiquitin ligases (UBR1 to UBR7) is a family that contains an evolutionally conserved UBR box domain, a substrate recognition domain [21,22]. This review discusses the structural features and signaling pathways mediated by these UBR box E3 ligases.

## 2. N-Degrons and the UBR Box E3 Ligases

According to the N-end rule, the lifespan of a protein depends on the character of its N-terminal residue. N-terminal residues that destabilize a protein are termed N-degrons, classified as type 1 or type 2. Type 1 N-degrons contain positively charged amino acids such as Arg, Lys, and His, and type 2 N-degrons include hydrophobic residues such as Phe, Trp, Tyr, Lue, and Ile. These N-degrons can be generated directly by nonprocessive proteases, including methionine–aminopeptidases (MetAPs), caspases, calpains, separases, or indirectly, by enzymatic cascades that mediate the post-translational arginylation of newly exposed Asn, Gln, Asp, Glu, and Cys in mammals [23,24,25,26,27,28,29,30]. Asn and Gln can be converted to Asp and Glu via deamidation mediated by the protein N-terminal asparagine amidohydrolase (NTAN1) and protein N-terminal glutamine amidohydrolase (NTAQ1), respectively [31,32,33]. N-terminal Cys can be oxidized by oxygen depletion or nitric oxide (NO) to become either Cys-sulfinic acid (CysO_2_H) or Cys-sulfonic acid (CysO_3_H) [34,35,36]. Recently, the formation of Cys-sulfinic acid has been shown to be mediated by cysteamine (2-aminoethanethiol) dioxygenase (ADO) [37]. N-terminal Asp, Glu, and oxidized Cys are conjugated with the amino acid L-Arg by arginyl-tRNA-protein transferase 1 (ATE1) to generate a canonical N-degron, Arg (Figure 1). Recently, some evidence has shown that stabilizing residues can also act as N-degrons in a context-dependent manner [38].

Seven UBR box E3 ligases have been identified in mammals (UBR 1-7) (Figure 2). The UBR box of UBR1, UBR2, UBR4, and UBR5 has been shown to bind type 1 N-degrons. In addition, UBR1 and UBR2 can bind type 2 N-degrons through an N domain present in both proteins [39,40]. However, UBR4 can also bind type 2 N-degrons, although no defined N domain has been identified [39,41]. The molecular mechanism by which UBR4 recognizes type 2 N-degrons requires further investigation. These N-degron-binding UBR box E3 ligases are termed N-recognins.

## 3. The Structure of the Family of UBR Box Proteins

In 1990, Varshavsky and colleagues identified UBR1 (ubiquitin–protein ligase E3 component N-recognin 1) whose molecular weight is around 225 kDa as the sole N-recognin in *Saccharomyces cerevisiae* [42]. Later, seven UBR box-containing proteins called UBR1 to UBR7 were identified in mammals. In addition to the UBR box, these proteins contain various domains and motifs seen in other E3 ubiquitin ligases such as E2 binding domains, RING, HECT, F-box, and PHD (Figure 2). Thus, they are classified as E3 ligases and serve as a platform through which ubiquitin can be transferred to the substrate.

### 3.1. UBR Box Protein Domains Associated with E3 Ubiquitin Ligases

UBR1, UBR2, and UBR3 are classified as RING-type E3s, constituting the majority of E3s. The RING domain mediates the direct transfer of ubiquitin from the E2 enzyme to the target protein and possesses conserved cysteine and histidine residues which bind two zinc ions, maintaining the overall structure [43,44,45,46]. Although the RING domain is responsible for recruiting ubiquitin-conjugating enzymes, its association with E2 does not always correlate with its E3 ligase activity. For example, in UBR1, the RING domain exhibits a low affinity for Ubc2 (an E2), and it is the BRR (basic rich region) domain in front of the RING domain which is required for the tight binding of Ubc2 to UBR1. However, the RING, not the BRR domain, is required for UBR1′s E3 ligase activity [47].

UBR5 belongs to the second-largest group of E3s, containing the HECT (homologous to E6-AP C-terminus) domain. The HECT domain consists of two different lobes, the N-terminal lobe (N-lobe) and C-terminal lobe (C-lobe), connected by a flexible linker. The N-lobe is responsible for binding the E2s, whereas the C-lobe bears the catalytic cysteine that accepts ubiquitin from the E2 to generate an E3–ubiquitin complex [48,49]. Therefore, unlike RING E3s, HECT E3s must be physically conjugated with ubiquitin before transferring the ubiquitin to the substrate. The HECT domain of UBR5 is present in its extreme C-terminus, and its catalytic cysteine is Cys2768 [48].

UBR6, also named FBXO11, is an F-box protein component of an SCF ubiquitin ligase consisting of Cullin 1 (CUL1), Skp1, RBX1, and the corresponding F-box protein [50,51]. CUL1 functions as a scaffold for the binding of SKP1/F-box protein complex on its N-terminus and binding the RBX1/E2 complex on its C-terminus. RBX1 recruits the E2 enzyme to the E3 ligase, whereas the F-box protein directly binds substrates to mediate ubiquitylation and proteasomal degradation, thus determining substrate specificity [52]. F-box proteins contain the F-box domain, a 40-amino-acid region which is required to bind Skp1 [53]. Accordingly, UBR6 is expected to act as an E3 ubiquitin ligase through this SCF complex.

UBR7 possesses the plant homeodomain (PHD) finger, a 50–80 amino acid protein domain with Cys and His residue patterns similar to that of the RING domain [45]. This domain was first reported in the Homeobox protein HAT3.1 of Arabidopsis in 1993 and is found in more than 100 human proteins [54]. Proteins with this domain are primarily involved in gene regulation in the nucleus [55]. Several proteins have been reported to bind to the methylated lysine of histone H3 through this domain [56]. Recently, UBR7’s PHD has been demonstrated to exhibit E3 ubiquitin ligase activity to monoubiquitinate histone H2B at lysine 120 (H2BK120Ub). The loss of UBR7 was correlated with the development of triple-negative breast cancer and metastatic tumors [57].

### 3.2. Substrate Recognition Domains of UBR Box-Containing N-Recognins

UBR boxes of N-recognins, including UBR1, UBR2, UBR4, and UBR5, are responsible for binding type-1 basic N-degrons. For the recognition of type-2 hydrophobic N-degrons, an N-domain present in UBR1 and UBR2 is required. The N-domain is a homologue of bacterial N-recognin, ClpS. In this section, the structural bases of substrate recognition by these domains are discussed.

#### 3.2.1. UBR Box

The crystal structures of the UBR box from human UBR1, human UBR2, and yeast UBR1 have been determined, greatly enhancing the understanding of the binding mechanism between the UBR box and substrates [58,59,60,61]. The UBR box possesses two zinc fingers that coordinate three zinc ions (Figure 3).

The UBR box has two pockets responsible for recognizing the first and the second residues of substrate N-degrons. The first pocket consists of D118, T120, F148, D150, and D153 of human UBR1 and forms a negatively charged surface where the first amino acid of type 1 N-degrons (Arg, Lys, or His) can bind (Figure 4A). T120, F148, and D150 mainly interact with the amino group of the first amino acid and the amide bond between the first and the second amino acids through charge–charge interactions. Therefore, the selectivity for the first amino acid is determined by the negatively charged surface formed by D118, T120, and D153. This surface forms adequate space where various positive residues can be bound. In this space, the ligand and a water molecule form a strong interaction through hydrogen bonding and charge–charge interaction with surface residues. Arg, with a large residue size and mono- and di-methylated Arg, fill the pocket with one water molecule (Figure 4B–D), whereas Lys and His bind to the surface residues with one or two water molecules (Figure 4E,F), respectively. However, they do not entirely fill the space; thus, they show lower affinity than Arg or modified Arg because the residue sizes are smaller [59].

There is a significant difference in the UBR box binding affinity for N-degrons with the same first amino acid but different second residues based on the interaction between the second pocket of the UBR box and the second amino acid of the N-degron. This pocket consists of five and six critical residues in the human and yeast UBR box, respectively (Figure 5A,B). The surface of this pocket is hydrophobic; therefore, most show high binding affinity when a hydrophobic residue is located at the second amino acid position (Figure 5C,D,J,K).

There are some differences between the structure and sequence of human and yeast UBR boxes. For example, R135 of yeast UBR1 corresponds to S111 of human UBR2, although they are entirely different in character and size. Furthermore, the T171 loop of yeast UBR1 is located close to the ligand-binding site, whereas the loop of human UBR1 is shorter and more distant from the binding site. These differences in the pocket surface cause differences in the substrate-binding mechanism.

When the RR-peptide binds to yeast UBR1, R135 moves backward to create a space, and the positive charge of the second Arg is stabilized together with a water molecule and D165 (Figure 5G,H). When Glu or Asp, which are negatively charged, are bound, the T171 loop is withdrawn, and a space for the negatively charged residue is formed (Figure 5E,F). The negatively charged Glu and Asp residues are stabilized with a water molecule immobilized by R135 and D165 (Figure 5I). On the other hand, in human UBR2, the glycine loop has been retracted, thus securing enough space for the second residue of the ligand to bind. Moreover, it is difficult to stabilize the positively charged residue of Arg because the R135 of yeast UBR1 is substituted to S111 of human UBR2 (Figure 5L). According to previous studies, as a result of measuring the affinity of yeast UBR1 and peptides, RRAA showed higher binding affinity (*K*_D_ = 17.7 ± 0.325 μM) than RDAA (*K*_D_ = 343.2 ± 36.4 μM). However, in the case of human UBR2, RDFS showed twice the thermal stability of RRFS, indicating that RDFS has a higher affinity than RRFS [59]. These results show that the amino acid sequences of the primary substrates of yeast UBR box and human UBR box are different.

#### 3.2.2. N-Domain

The N-domain is found only in UBR1 and UBR2 and is responsible for recognizing type 2 N-degrons, including Phe, Trp, Tyr, Leu, and Ile [39,40]. This N-domain is a homologue of ClpS that recognizes type 2 N-degrons in bacteria (Figure 6A) [62]. The structure of ClpS is well defined and has a deep, narrow, hydrophobic pocket for N-degron binding (Figure 6B). The entrance of this pocket has a negatively charged surface composed of N34, D35, T38, M40, and H65 (Figure 6D). The N34 and H66 residues and oxygen atoms of D35, T38, and M40 amide bonds form a charge–charge interaction with the ligand’s amino group and amide bond. The inside of the pocket comprises hydrophobic residues, consisting of hydrophobic residues V43, V65, and L109, which stabilize the first residue of type 2 N-degrons (Figure 6F) [63,64]. Homology modeling of the human UBR1 N-domain based on the ClpS structure of *E. coli* (Figure 6C) showed that the sequence identity between ClpS and N-domain is 18%, and the similarity is 36%. Despite the low similarity, all of the residues forming the pocket are well conserved, except D263 of the N-domain, corresponding to H66 of ClpS (Figure 6E,G).

## 4. Signaling Pathways Controlled by UBR Box N-Recognins in Mammals

The N-recognin functions of UBR1, UBR2, UBR4, and UBR5 have been well characterized; however, the functions of UBR3, UBR6, and UBR7 remain largely unknown. This section discusses the role of these UBR N-recognins in signaling pathways, including G-protein signaling, apoptosis, mitochondrial quality control, inflammation, and DNA damage signaling.

### 4.1. G-Protein Signaling Pathway

G-protein-coupled receptors (GPCRs) characterized by seven-(pass)-transmembrane domains are the largest and the most diverse group of membrane receptors in eukaryotes [67,68,69]. GPCRs are associated with G-proteins, which are heterotrimeric proteins comprising subunits, α, β and γ [70,71]. Upon activation by extracellular ligands or signal mediators [72,73], the GPCRs undergo a conformational change inducing guanine–nucleotide exchange factors (GEFs) to catalyze the exchange of GDP bound to the G〈 subunit to GTP [74,75,76]. This results in the dissociation of the GTP-bound Gα subunit from the Gβγ dimer (Figure 7) [77,78,79]. Both diffuse throughout the membrane to turn on diverse intracellular signal transduction pathways by interacting with other membrane proteins. Turning off G-protein signaling requires the hydrolysis of the GTP-bound G〈 subunit to GDP via its intrinsic GTPase activity. The GTPase activity of the G〈 subunit is weak; therefore, augmentation by GTPase-activating proteins (GAPs), also called G-protein signaling regulators (RGSs), is required to shut down G-protein signaling efficiently [80,81,82,83].

Amongst RGS family proteins, RGS4, RGS5, and RGS16 are known physiological substrates of the Arg/N-degron pathway [34,36,84]. The first methionine residue of RGS4, RGS5, and RGS16 is constantly removed by methionine aminopeptidases (MetAPs), exposing the second cysteine residue, which can be oxidized by reactive oxygen species (ROSs) or cysteamine (2-aminoethanethiol) dioxygenase (ADO) and, in turn, arginylated by arginyltransferase 1 (ATE1) under normoxia [36,37]. The N-terminal Arg of the arginylated RGS proteins is recognized by the UBR box of UBR1/2, which results in poly-ubiquitination and proteasomal degradation. Accordingly, knocking out UBR1/2 has been shown to stabilize RGS4 and RGS5 and exhibited the impairment of neurodevelopment and cardiovascular development in mice, suggesting the importance of UBR1/2′s N-recognin function in controlling G-protein signaling-mediated biological processes [34,36,85,86,87] (Figure 7). Therefore, these RGS proteins have a very transient existence under normoxia due to protein degradation mediated by UBR1/2, prolonging G-protein signaling. However, the stabilization of RGS proteins under hypoxia or the impairment of UBR1/2 restricts G-protein signaling [36].

### 4.2. Apoptosis Signaling Pathway

Apoptosis is a type of programmed cell death utilized by multicellular organisms to selectively eliminate damaged or abnormal cells to maintain homeostasis [88,89]. The dysregulation of apoptosis leads to various diseases such as cancer and neurodegenerative diseases [90,91,92]. Studies have shown that UBR E3 box ligases, in their function as N-recognins of the N-degron pathway, play a role in the negative regulation of apoptosis [28,93,94].

Apoptosis is characterized by the activation of numerous proteases such as caspases and calpains, responsible for the cleavage of over 1000 cellular proteins [95,96,97]. This protease activity generates numerous protein fragments, some of which, termed pro-apoptotic fragments, promote further apoptotic activity in a positive feedback system. Many of these pro-apoptotic fragments have acquired N-degrons (Figure 8) directly from proteolytic cleavage or through the actions of ATE 1, which are recognized by E3 N-recognins and removed via the UPS. Selective removal of pro-apoptotic fragments inhibits apoptosis and promotes cell survival [28,29].

The evidence for the apoptosis inhibitory effect of the UBR box E3 ligases comes from multiple studies. Using colony-forming and TUNEL assays, it was observed that apoptotic cell death was significantly increased in ATE1 or UBR1/2-deficient cells under apoptotic stimuli such as UV irradiation, staurosporine, and TNF-α [28]. According to another report, the depletion of UBR1, UBR2, UBR4, and UBR5 using small interfering RNA (siRNA) increased apoptosis in various cancer cells [98]. A study of RIPK1, a caspase-8 target, showed that its ubiquitination promotes cell survival [99]. On the other hand, RIPK1 cleavage by caspase-8 generates a pro-apoptotic C-terminal fragment, Cys325-RIPK1, containing the death domain of RIPK1. However, this fragment can be degraded by the Arg/N-degron pathway. When cysteine exposed at the N-terminus of RIPK1 is replaced with a stabilizing residue, valine, metabolic stabilization occurs in which fragments are accumulated in the cytoplasm without degradation. Metabolic stabilized Val325-RIPK1 promotes cell death by significantly increasing caspase-3 activity [28]. Therefore, the metabolic stabilization of the RIPK1 C-terminal fragment increases hypersensitivity to programmed cell death. Interestingly, RIPK1 is one of the upstream regulators leading to necroptosis [100]. When an N-degron pathway inhibitor called RFC11 [101] was cotreated with the anticancer drug shikonin in CT26 colon cancer cells, RIPK1 stability was increased, thereby significantly reducing cell viability and tumor growth through necroptosis induction [94]. ETK/BMX is a Tec non-receptor tyrosine kinase family member, and is also a target of caspases [93]. This kinase is involved in cell survival in response to radiation-induced apoptosis in prostate cancer and breast cancer and modulates pro-apoptotic functions [102,103,104,105]. BMX kinase becomes sensitive to apoptotic signaling via the C-terminal fragment generated through caspase cleavage in prostate-cancer-derived PC3 cells. The Trp243-BMX Ct-fragment has a destabilizing residue, tryptophan, at the N-terminus, which is recognized by UBR1/2 and rapidly removed through the UPS. However, when Trp is substituted with Val, this fragment is stabilized, promoting apoptotic cell death in PC3 cells. Intriguingly, phosphorylation of Tyr566 of this fragment is known to inhibit the Arg/N-degron pathway-mediated degradation [93].

Therefore, UBR box E3 ligases can inhibit apoptosis through the clearance of pro-apoptotic fragments via the UPS. However, there is a report that activated caspases can functionally inhibit ATE1 and UBR1, which both contain caspase-8 cleavage sites. In the case of ATE1, it was confirmed that the function of R-transferase was significantly reduced by caspase-8. Accordingly, when the caspase-8 activity was inhibited by treatment with the pan-caspase inhibitor Z-VAD-FMK, the R-transferase function of ATE1 was restored [28].

### 4.3. Mitochondrial Quality Control Pathway

Neurodegenerative diseases are caused by protein aggregation and can also be generated from a failure of mitochondrial quality control [106]. Mitochondrial dysfunction increases oxidative stress and affects various cellular signaling pathways, leading to neuronal cell death, associated with neurodegenerative conditions such as Parkinson’s disease. PINK1 and Parkin play an essential role in mitochondrial quality control, and their mutations cause familial Parkinson’s disease [106,107,108].

PINK1 undergoes rapid and continuous degradation in normal mitochondria. Under normal mitochondrial conditions, the N-terminus of PINK1 is inserted into the inner mitochondrial membrane (IMM) via TOM and TIMM23 translocator complexes, where PARL, an IMM protease, cuts PINK1 between residues Ala103 and Phe104, releasing a C-terminal fragment of PINK1 (Phe104-PINK1) into the cytosol [109,110,111,112,113,114]. This fragment is recognized by UBR1, UBR2, and UBR4 and removed through the UPS (Figure 9A) [112]. However, in dysfunctional mitochondria, PINK1 accumulates in the outer mitochondrial membrane (OMM). Accumulated PINK1 recruits Parkin, an E3 ubiquitin ligase, to the target mitochondria, where it ubiquitinates substrates present in the OMM, leading to the removal of the dysfunctional mitochondria by mitophagy (Figure 9B) [108,115,116]. The biological consequences of UBR box N-recognin impairment on the proteasomal degradation of the PINK1 fragment need further investigation.

### 4.4. Inflammatory Signaling Pathways

Inflammation is a protective response induced by the evolutionarily conserved innate immune system to fight against harmful stimuli such as pathogens, damaged cells, or irritants [117]. The innate immune system is initiated by the recognition of pathogen-associated molecular patterns (PAMPs) and danger-associated molecular patterns (DAMPs) by pattern-recognition receptors (PRRs) as the first defense mechanism of our immune system [118]. When PAMPs or DAMPs are recognized, several PRRs form large multiprotein complexes called inflammasomes, which regulate the activity of caspase-1. Inflammasome-activated caspase-1 promotes the secretion of the proinflammatory cytokines such as IL-1β and IL-18 and activates pore-forming protein gasdermin D (GSDMD) [119,120,121,122]. In addition, these proteins promote cell death, which is called pyroptosis.

PRRs are classified into subfamilies such as Toll-like receptor (TLR), C-type lectin receptor (CLR), and NOD-like receptor (NLR), according to their location and domain composition [123]. It is known that NLRP1, a member of the NLR subfamily, is regulated by the Arg/N-degron pathway [124,125]. NLRP1 has C-terminal FIIND and CARD domains and NACHT and LRR domains, specific domains of the NLR family [126]. When the CARD domain of this protein is exposed, caspase-1 is activated by CARD oligomerization and pyroptosis occurs, activating inflammasomes. The NLRP1 protein contains two cleavage sites whose cleavage is required for the exposed CARD domain inflammatory activity. The mechanism of NLRP1 inflammasome activation by N-terminal and C-terminal fragments produced by this cleavage is well known through studies of mNLRP1B, a mouse NLRP1 [124,125,126,127].

mNLRP1B contains an autocleavage site in the FIIND domain, which consists of the ZU5 and UPA subdomains. N-terminal and C-terminal fragments are generated when mNLRP1B is auto-cleaved between ZU5 and UPA (Phe983-Ser984) [128,129,130]. However, this single cleavage does not result in an inflammatory response due to the autoinhibitory activity of the N-terminal fragment of the protein. Many pathogens utilize mechanisms such as anthrax lethal factor (LF), a metalloprotease and component of anthrax lethal toxin (LT), to target and destroy NLRP proteins in an attempt to evade an immune response. However, when LF enters the cytosol and cleaves mNLRP1B between Lys44 and leu45, it removes the autoinhibitory effect of mNLRP1B and exposes the CARD domain of the C-terminal fragment, inducing pyroptosis and inflammation [128,130,131,132,133] by generating an N-terminal fragment (Leu45-mNLRP1B-F983) containing an N-degron which is recognized by N-recognins such as UBR2 and UBR4 and degraded by the UPS (Figure 10A) [124,125]. Accordingly, N-recognins UBR2 and UBR4 were identified through genome-wide CRISPR-Cas9 screening to find proteins related to LT-induced NLRP1 inflammasome activity. Moreover, degradation of the N-terminal fragment of mNLRP1B was significantly reduced due to the deficiency of UBR2 and UBR4 in RAW264.8 cells, and the resistance to LT-induced pyroptosis [124,125].

In contrast to the NLRP1 pathway, there is a report that inflammation can be alleviated by the Arg/N-degron pathway [134]. PAMPs and DAMPs initiate inflammatory responses, and various proinflammatory fragments are generated by activated inflammatory caspases or other proteases involved in immune responses [118,134]. Some of these proinflammatory fragments contain destabilizing N-terminal residues. These include caspase-generated Asn120-CASP1, Gln81-CASP4, Gln139-CASP5, and Cys149-Rab39, as well as Ile29-GRZA and Ile27-GRZM produced by endopeptidase DPP1 (Figure 10B). These fragments are substrates of N-recognins, such as UBR1, UBR2, UBR4, and UBR5, and degraded via the UPS. Indeed, when the UBR1, UBR2, UBR4, and UBR5 were knocked down using RNAi, the secretion of IL-1β was significantly increased [134], suggesting that the degradation of these proinflammatory fragments via the Arg/N-degron pathway plays a vital regulatory role in inflammatory responses.

### 4.5. DNA Damage Response Pathway

Replication stress, defined as the slowing or stalling of replication fork progression and DNA synthesis, can cause DNA mutations and chromosomal aberrations [135,136,137]. Thus, failure to counteract genotoxic threats can lead to cancer, developmental disorders, ciliopathies, and laminopathies [136,138,139,140]. Various well-known endogenous and exogenous sources of DNA damage, such as oxidation, chemical mutagens, and ultraviolet radiation, can interfere with the proper progression and completion of the replication process, resulting in genome instability [141,142]. An essential factor of the replication machinery is the proliferating cell nuclear antigen (PCNA) which plays an essential role in maintaining genomic integrity and promoting DNA replication by guiding replicative DNA polymerases at replication forks [143,144]. UV irradiation and various DNA-damaging agents (MMS, mitomycin C, cisplatin, and H_2_O_2_) lead to the stalling of the replication fork and the release of DNA polymerase from PCNA [142,145,146,147,148,149]. As part of the DNA damage tolerance mechanism, RAD18, a ubiquitin ligase, can monoubiquitinate PCNA, preventing replication fork collapse that can trigger cell death or genome instability by enabling the DNA replication of damaged templates through translesion synthesis (TLS) [150,151]. According to recent studies, SDE2 protein is implicated in genome instability caused by replication stress by modulating this mechanism [152,153].

SDE2, a genome surveillance protein, is a highly conserved human protein that possesses a DNA-binding SAP (SAF-A/B, Acinus, and PIAS) domain, which is frequently found in DNA repair proteins such as PIAS1, Ku70, and RAD18 [152,154]. In addition, SDE2 also contains a ubiquitin-like (UBL) domain on its N-terminus, responsible for interacting with PCNA. Under replication stress conditions, SDE2 is recruited to the replication fork, where it binds to the DNA through its SAP domain and interacts with PCNA via its PIP box in UBL domain [152]. The deubiquitinating enzyme (DUB) cleaves SDE2, resulting in a C-terminal fragment, Lys78-SDE2, which inhibits the monoubiquitination of PCNA, impairing S phase progression. Lys78-SDE2 at the replication fork must be degraded to overcome the replication stress via the PCNA-dependent DNA damage bypass. Lys78-SDE2 contains a canonical N-degron, which is recognized by UBR1/2 and ubiquitinated. However, to be degraded, Lys78-SDE2 must also be phosphorylated by ATR kinase. The phosphorylation of Lys78-SDE2 recruits the p97^UFD1-NPL4^ segregase complex, resulting in the release of ubiquitinated Lys78-SDE2 from the chromatin and allowing for its degradation via the UPS, thus promoting stalled fork recovery and S phase progression [153]. Therefore, UBR1/2 plays a crucial role in fork recovery, DNA replication, and S phase progression under replication stress conditions (Figure 11).

## 5. Concluding Remarks

UBR box E3 ligases including UBR1, UBR2, UBR4, and UBR5 are the primary N-recognins of the N-degron pathways. The mechanisms through which some of these N-recognins bind their substrates has been well characterized by structural analysis, as detailed in this review. These N-recognins have a wide variety of physiological substrates (Table 1). The interactions of these N-recognins with these substrates play essential regulatory roles in many signaling pathways such as G-protein signaling, apoptosis, inflammation, mitochondrial quality control, and replication stress. The dysregulation of these pathways often leads to disease conditions such as cancer and neurodegeneration. Thus, understanding the mechanism mediated by these N-recognins in these pathways may provide crucial therapeutic targets for future drugs.

## 6. Methods

The homology model was generated by the SWISS-MODEL program [65,66]. The all-3D structure models such as stick, cartoon, and surface models were generated by the PyMol [155].

## Figures and Tables

**Figure 1 ijms-22-08323-f001:**
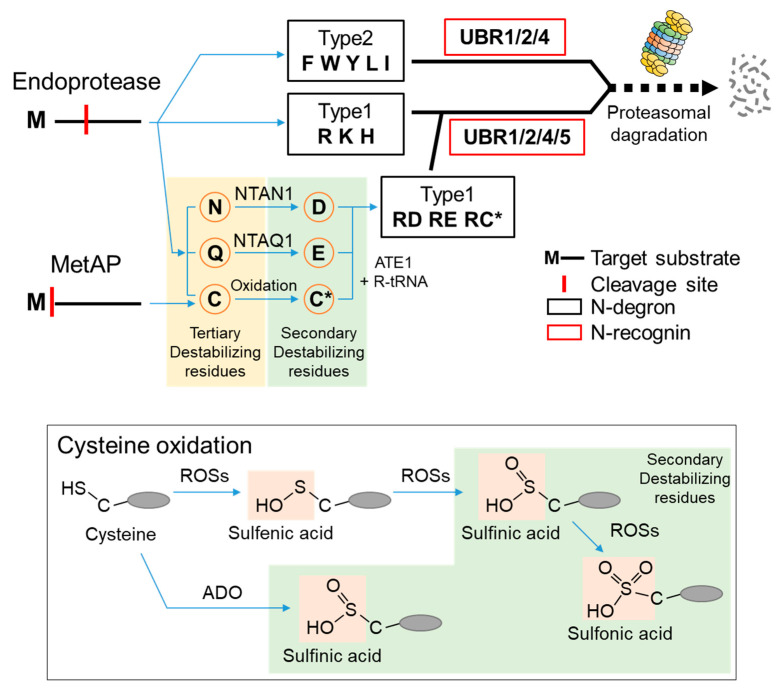
The classical Arg/N-degron pathway. N-degrons are classified as type 1 or type 2 according to the residues exposed by proteases such as Endoprotease and MetAP. Asn, Gln, and Cys are tertiary destabilizing residues (in the light-yellow box), converted into secondary destabilizing residues, Asp, Glu, and oxidized-Cys (in the light-green boxes), respectively, and finally become type 1 Arg N-degrons through ATE1-mediated arginylation (C* denotes the oxidized N-terminal cysteine residue). Cysteine is oxidized by typical ROSs and the recently reported ADO (ROSs, reactive oxygen species; ADO, cysteamine (2-aminoethanethiol) dioxygenase).

**Figure 2 ijms-22-08323-f002:**
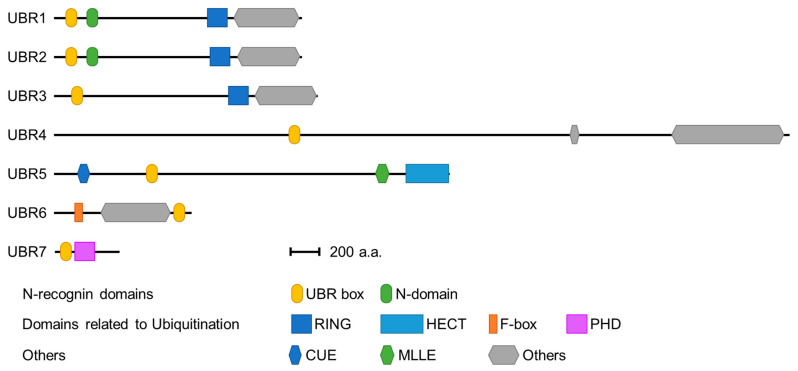
Domains of the UBR box protein family. All UBR proteins have a UBR box (yellow ellipse) to recognize N-degrons, and this UBR box is the signature of the UBR family. UBR1 and UBR2 also have an N-domain (green ellipse) that recognizes the type 2 N-degrons. In addition, these UBR proteins have a RING (navy blue square), HECT (blue square), F-box (orange box), or PHD domain (purple box) for E2 binding or ubiquitin conjugation. Other domains include the CUE domain, which recognizes ubiquitin, and the MLLE domain, known to regulate the catalytic activity of HECT.

**Figure 3 ijms-22-08323-f003:**
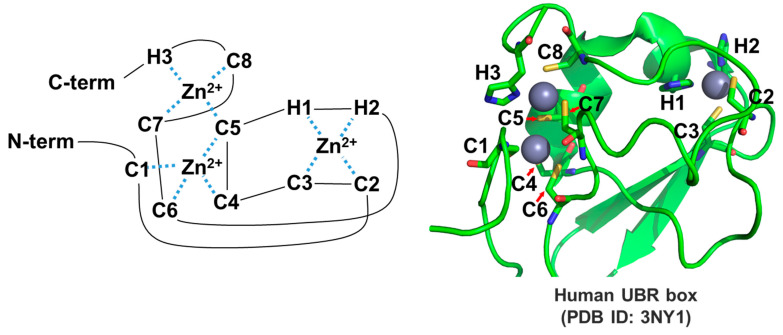
Three-dimensional structures of the UBR box. The UBR box coordinates three zinc ions. Two cysteines (C2 and C3) and two histidines (H1 and H2) coordinate one zinc. Six cysteines (C1, C4, C5, C6, C7, and C8) and one histidine (H3) coordinate the other two zincs.

**Figure 4 ijms-22-08323-f004:**
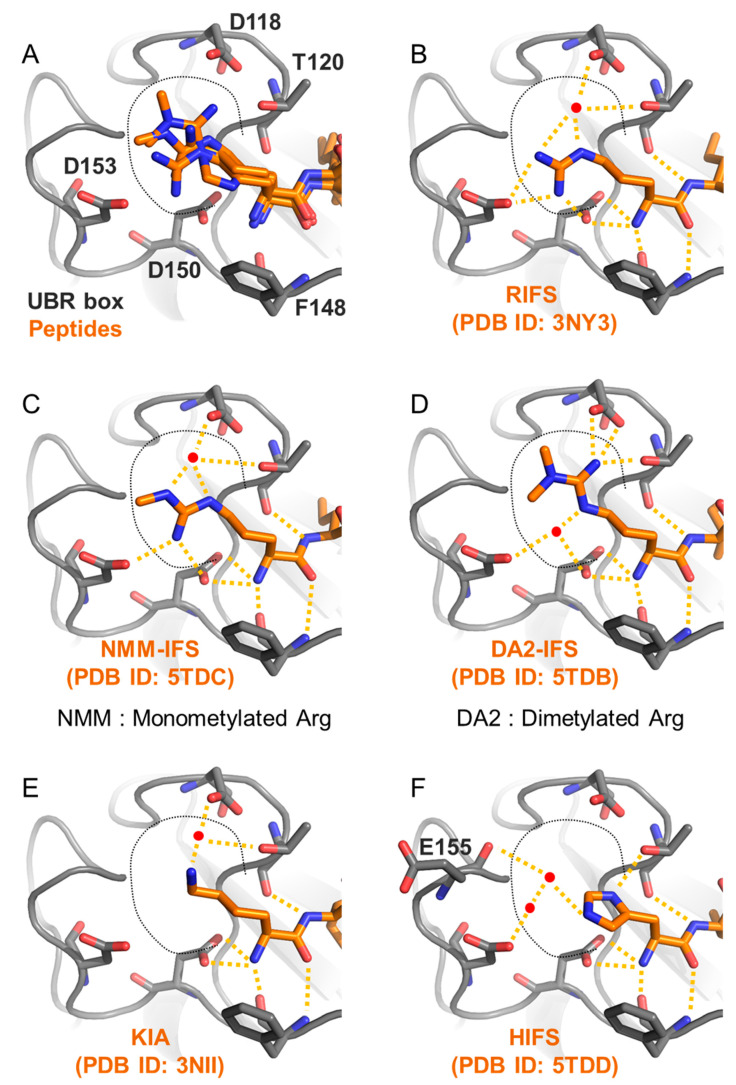
The first pocket of the UBR box recognizes the first residue of type 1 N-degrons. (**A**) Residues constituting the first pocket have enough space (black dotted line) for positively charged residues to enter. (**B**–**F**) Hydrogen bonding and charge–charge interactions (yellow dotted lines) generated when positively charged residues and modified arginine residues bind to the pockets, and water molecules (red dots) fill the empty spaces.

**Figure 5 ijms-22-08323-f005:**
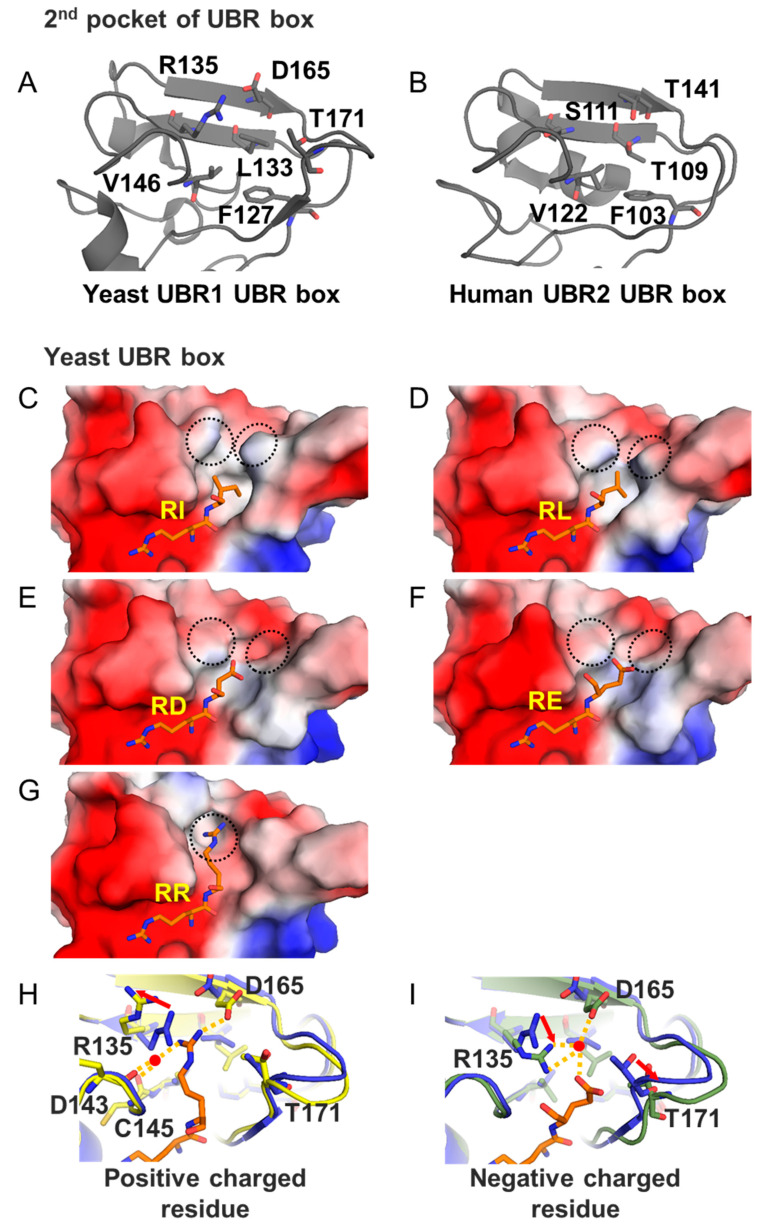
The second pocket of the UBR box recognizes the second residue of type 1 N-degron. (**A**,**B**) Residues constituting the second pocket of the UBR boxes of yeast UBR1 and human UBR2 are represented by sticks (PDB ID 3NIH and 3NY3). (**C**,**D**) These surface models interact with hydrophobic residues (Ile and Leu). Blue on the surface means a positively charged, and red means a negatively charged. The ligands are bound to the hydrophobic surface of this pocket (PDB ID 3NIH and 3NIN). (**E**,**F**) Acidic residues (Asp and Glu) are bound to the UBR box. T171 is pushed back by the negatively charged of acidic residues (PDB ID 3NIL and 3NIK). (**G**) The basic residue Arg is bound to the UBR box. It can be seen that R135 is pulled back to bind the large and positively charged Arg residue (PDB ID 3NIM). (**H**,**I**) The changes in the secondary pocket that occur when charged residues bind are represented by cartoon and stick models (PDB ID 3NIM and 3NIK). (**J**,**K**) The ligand structures bound to the human UBR box are shown as surface and stick models with a larger surface than yeast because residues such as R135 and T171 do not exist in the human UBR box (PDB ID 3NY3 and 5TDA). (**L**) The secondary pocket of the human UBR box and its ligands are shown as cartoon and stick model (PDB ID 3NY3 and 5TDA).

**Figure 6 ijms-22-08323-f006:**
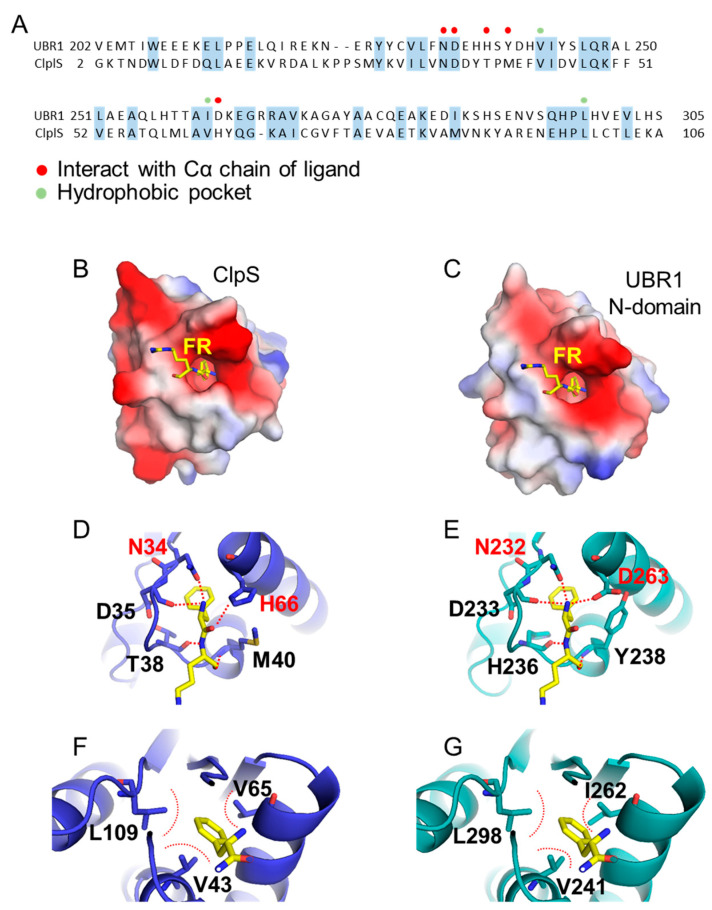
Homology models of the human UBR1 N-domain based on ClpS of *Escherichia coli*. The homology model of the N-domain of human UBR1 was generated based on the ClpS structure (PDB ID 2W9R) using the SWISS-MODEL program [65,66]. (**A**) The amino acid sequences of human UBR1 and *E. coli* ClpS are aligned. Conserved residues are highlighted with light blue boxes. Red and light green dots indicate the residues recognizing type 2 N-degrons. (**B**,**C**) The surface models of the ClpS and N-domain are colored by charge. The ligand, Phe–Arg peptide, is represented as a stick. (**D**,**E**) The residues constituting the entrance of the hydrophobic pocket interact with the backbone of the first amino acid (Phe) by charge–charge interactions and hydrogen bonding (red dotted lines). (**F**,**G**) The hydrophobic residue of the first amino acid is bound to the hydrophobic pocket. The red dotted lines mark the space created by the hydrophobic residues constituting the pocket.

**Figure 7 ijms-22-08323-f007:**
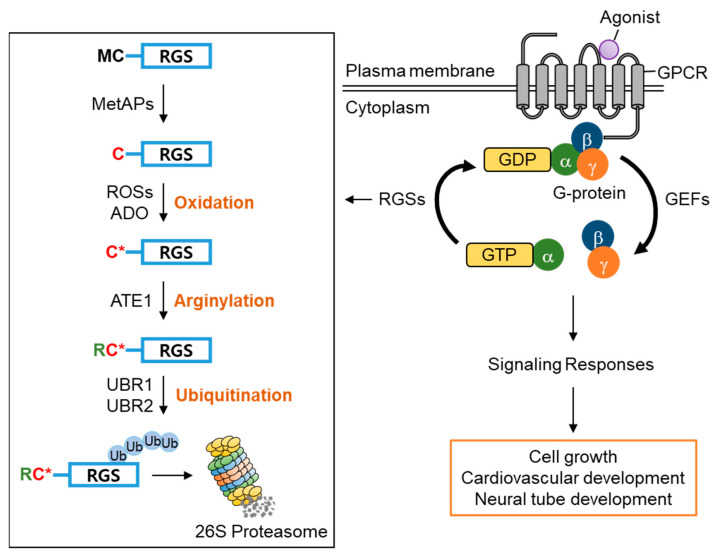
A model describing the regulatory role of UBR1 and UBR2 in G-protein signaling pathways via mediating RGS protein degradation. In GPCRs (G-protein-coupled receptors), heterotrimeric G-proteins are dissociated into G〈 and G®© subunits by external ligands or signal mediators. Activated GTP-bound G〈 and G®© stimulate the downstream signaling pathway associated with cell growth and cardiovascular development. The proper regulation of GTP-Gα activity by GTPase-activating RGS proteins is vital in the GPCR-related signaling pathway. Among the RGS proteins, RGS4, RGS5, and RGS16 are cleaved by MetAP to expose a cysteine residue at the N-terminus. After which, these RGS proteins undergo oxidation, followed by arginylation (C* denotes the oxidized N-terminal Cysteine residue). Arginylated RGS proteins are recognized by UBR1 and UBR2 for ubiquitination and degradation. When the Arg/N-degron pathway is genetically inhibited, metabolically stabilized RGS proteins promote the hydrolysis of the GTP-bounded G〈 subunit, which leads to inactivation of the GPCR signaling pathway.

**Figure 8 ijms-22-08323-f008:**
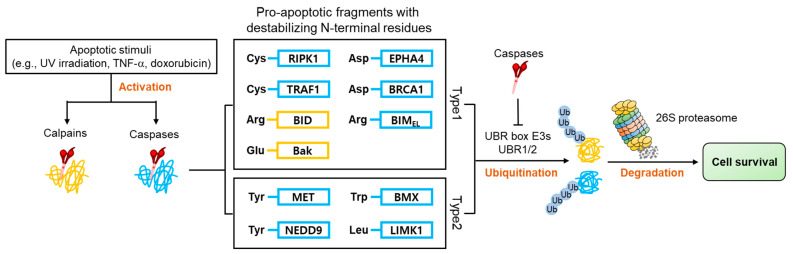
A model depicting the degradation of pro-apoptotic fragments mediated by UBR1/2 in response to apoptotic stimuli. Caspases or calpains generate pro-apoptotic fragments during the induction of apoptosis. These proteins expose destabilizing residues at the N-terminus, which are short-lived N-degron substrates. As a result, these pro-apoptotic fragments are selectively degraded by the Arg/N-degron pathway, contributing to cell survival. As a negative feedback mechanism, caspases can also inhibit UBR1 function by inducing its cleavage.

**Figure 9 ijms-22-08323-f009:**
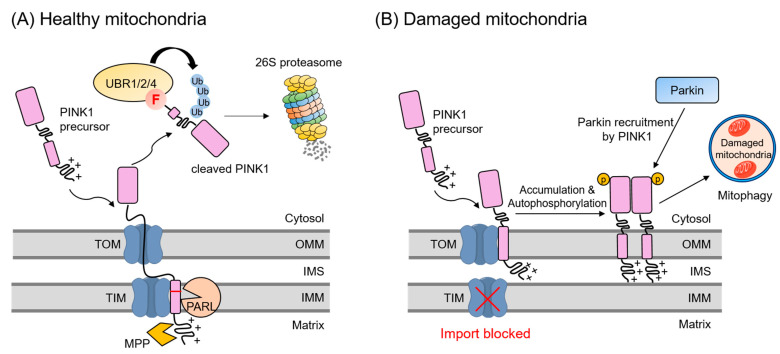
A schematic model describing mitochondrial quality control mediated by UBR1/2/4 in response to the mitochondrial damage. (**A**) PINK1 is continuously imported into healthy mitochondria through TOM and TIM complexes under steady-state conditions. After which, the precursor PINK1 is cleaved by MPP and PARL proteases, respectively. PARL cleaves between Ala103 and Phe104 of PINK1 to expose phenylalanine, a known type-2 destabilizing residue of the Arg/N-degron pathway, at the N-terminus. The cleaved PINK1 is released into the cytosol. Phe104-PINK is recognized by UBR1, UBR2, and UBR4 for ubiquitination and proteasomal degradation. (**B**) When depolarization or mitochondrial import is blocked, PINK1 accumulates on the OMM. The accumulation and autophosphorylation of PINK1 recruit Parkin E3 ligase from the cytosol to the mitochondria, leading to the degradation of damaged mitochondria through mitophagy.

**Figure 10 ijms-22-08323-f010:**
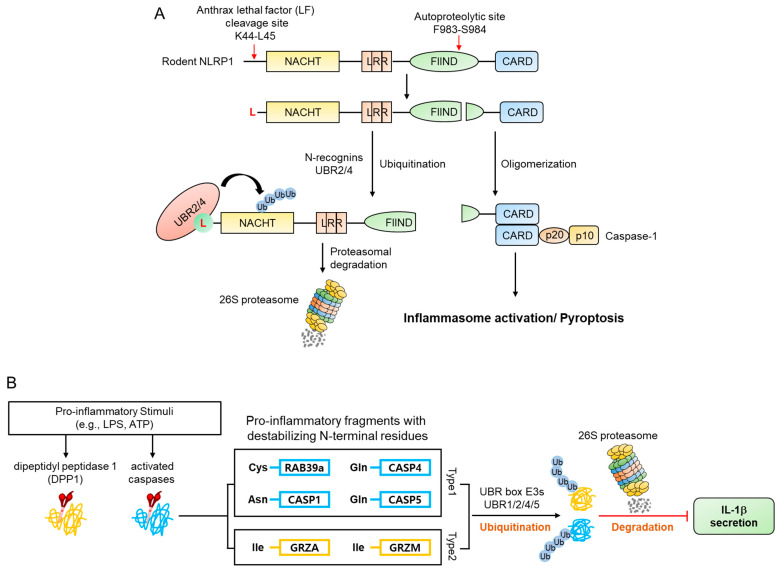
Schematic diagrams describing the regulation of inflammation by UBR box N-recognins. (**A**) A model depicting LT-induced mNLRPB inflammasome activation by UBR2/4. The anthrax lethal factor (LF) is known to induce mNLRP1B inflammasome activation and pyroptosis. Although the detailed mechanism of LT-induced mNLRP1B inflammasome activation has not been elucidated, it has recently been shown that the Arg/N-degron pathway is involved in mNLRP1B inflammasome activation. LF directly cleaves mNLRP1B to generate an N-terminal fragment and a C-terminal fragment. Despite cleavage into N-terminal and C-terminal fragments by LF, mNLRP1B remains autoinhibited. To activate the inflammasome, the N-terminal fragment of mNLRP1B is degraded in an Arg/N-degron pathway-dependent manner by UBR2 and UBR4, as identified through CRISPR-Cas9 screening. This process releases the CARD domain-containing C-terminal fragment of mNLRP1B and induces pyroptosis through interaction with caspase-1. (**B**) A model depicting the degradation of proinflammatory fragments mediated by UBR1/2/4/5. Potential proinflammatory Arg/N-degron substrates are generated by activated inflammatory caspases and several other proteases under inflammatory stimuli such as LPS. The resulting N-degron substrates Cys-RAB39a, Asn-CASP1, Gln-CASP4, CASP5, Ile-GRZA, and Ile-GRZM are generated by activated caspases, autoprocessing, or by endopeptidases such as DPP1. These fragments, which can cause an inflammatory response, expose destabilizing residues at the N-terminus. These fragments are recognized by N-recognins and then degraded through 26S proteasome, as evidenced by the depletion of UBR1, UBR2, UBR4, and UBR5, causing a significant reduction in LPS-induced IL-1β secretion.

**Figure 11 ijms-22-08323-f011:**
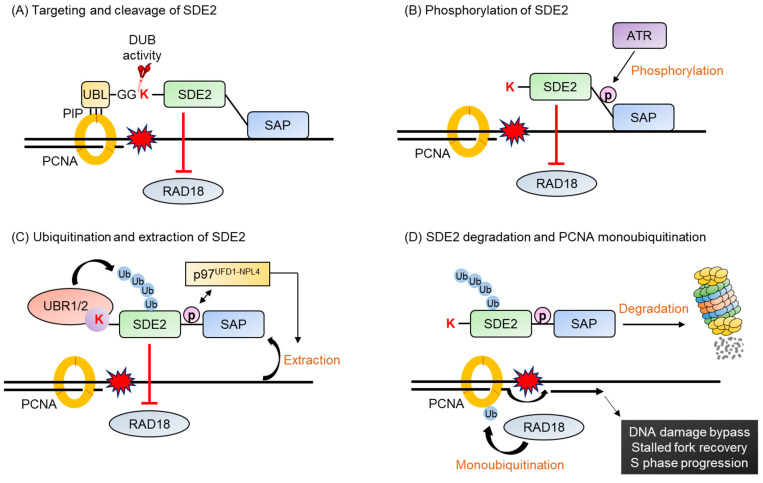
UBR1/2 mediated SDE2’s C-terminal fragment degradation upon replication stress, leading to recovering a stalled replication fork. The irradiation of cells with ultraviolet C, one of the causes of DNA replication stress, results in DNA lesions that block replication. The Arg/N-degron pathway is involved in counteracting DNA replication stress. Monoubiquitination of PCNA plays a vital role in coordinating DNA repair against replication-blocked lesions by providing a platform to recruit factors necessary for DNA repair. To counteract DNA replication stress, the C-terminal fragment of SDE2 needs to be degraded (**A**). Under UV-induced replication stress, SDE2 is targeted to the replication fork by interacting with PCNA through the PIP box of the UBL domain. After which, a C-terminal fragment of SDE2 (SDECt) is generated by the cleavage of the diglycine motif by DUB. Intriguingly, SDECt inhibits UV damage-inducible PCNA monoubiquitination by RAD18 ubiquitin E3 ligase. (**B**) Damage-inducible SDE2 Ct phosphorylation of Ser266 or Thr319, or both by ATR. (**C**) SDE2Ct, which has an N-terminal lysine, is recognized by UBR1 and UBR2 for polyubiquitination. In addition, phosphorylated SDE2Ct facilitates the interaction of p97^UFD1-NPL4^ and enables the extraction of ubiquitinated SDE2Ct from chromatin. (**D**) Consequently, degradation of SDECt by the Arg/N-degron pathway-ATR- p97^UFD1-NPL4^ axis promotes the monoubiquitination of PCNA by RAD18 E3 ligase, leading to DNA damage bypass, stalled fork recovery, and S phase progression.

**Table 1 ijms-22-08323-t001:** Physiological N-degron substrates of the Arg/N-degron pathway.

Biological Function	Species	Substrate	Pro-N-degron	N-degron	Modifications	Ref.
G-protein signaling	*Mus musculus*	RGS4	Cys2	Arg-Cys*	MetAPs cleavage, oxidation, arginylation	[34,36,84]
	*Mus musculus*	RGS5	Cys2	Arg-Cys*		
	*Mus musculus*	RGS16	Cys2	Arg-Cys*		
Apoptosis	*Mus musculus*	RIPK1	Cys325	Arg-Cys*	Endoproteolytic cleavage by caspase, oxidation, arginylation	[28,30,93]
	*Mus musculus*	TRAF1	Cys157	Arg-Cys*		
	*Mus musculus*	BRCA1	Asp1119	Arg-Asp	Endoproteolytic cleavage by caspase, arginylation	
	*Mus musculus*	EPHA4	Asp774	Arg-Asp		
	*Mus musculus*	BIM_EL_	-	Arg12	Endoproteolytic cleavage by caspase	
	*Mus musculus*	MET	-	Tyr1001		
	*Mus musculus*	NEDD9	-	Tyr631		
	*Homo sapiens*	LIMK1	-	Leu241		
	*Homo sapiens*	BMX	-	Trp243		
	*Homo sapiens*	BID	-	Arg71	Endoproteolytic cleavage by calpain, arginylation	[29]
	*Homo sapiens*	Bak	Glu16	Arg-Glu		
Mitochondrial quality control	*Homo sapiens*	PINK1	-	Phe104	Endoproteolytic cleavage by PARL	[112]
mNLRP1B Inflammasome	*Mus musculus*	NLRP1B	-	Leu45	Endoproteolytic cleavage by anthrax lethal factor	[124,125]
Inflammatory response	*Homo sapiens*	Caspase-1	Asn120	Arg-Asp	Auto-cleavage, deamidation, arginylation	[134]
	*Homo sapiens*	Caspase-4	Gln81	Arg-Glu		
	*Homo sapiens*	Caspase-5	Gln138	Arg-Glu		
	*Homo sapiens*	RAB39a	Cys149	Arg-Cys*	Endoproteolytic cleavage by caspase-1, oxidation, arginylation	
	*Mus musculus*	Granzyme A	-	Ile29	Endoproteolytic cleavage by DPP1	
	*Mus musculus*	Granzyme M	-	Ile27		
Genome stability	*Homo sapiens*	SDE2	-	Lys78	Endoproteolytic cleavage by DUB	[152,153]

MetAP. Methionine aminopeptidases; Cys *, oxidized cysteine; DUB, deubiquitylating enzyme; PARL, presenilin-associated rhomboid-like protein. N-degrons are expressed in red.

## Data Availability

Data available in a publicly accessible repository.

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
