# Peer review of "Signaling Pathways Regulated by UBR Box-Containing E3 Ligases"

_ijms, 2021, doi:10.3390/ijms22158323_

Round 1

Reviewer 1 Report

The manuscript "Signaling pathways regulated by UBR box-containing E3 ligases." by Kim et al. is a review about the structure, activity and signaling function of UBR box ligases.

The review is very informative and the 11 figures are of high quality. Therefore, I suggest to accept the manuscript in the present form.

Author Response

Thank you very much for your time to review our manuscript.

Reviewer 2 Report

The topic of the review titled "Signaling pathways regulated by UBR box-containing E3 ligases" by Kim et al. concerns a peculiar class of E3 ubiquitin ligases, namely UBR E3 ligases. More in detail, in their manuscript, the authors first describe some general and structural features of the UBR proteins and physical interactions with their respective substrates, namely N-degrons. Then in the second part they provide and overview on the roles played by UBR in few signaling pathways ranging from G-protein coupled receptors, programmed cell death, inflammation and eventually DNA-damage recognition.   On the whole, I appreciated the review very much. Indeed, it is well articulated and clearly written.   There are only few minor issues that need to be fulfilled prior publication mostly concerning oversights and typos. Please find below few examples 1. Figure 1 and that of the Figure 7, please indicate the meaning of the asterisk. 2. It would be appreciable add a scale bar to the Figure 2 (e.g. 1cm=200 aa). 3. A couple of typos need to be edited (e.g. page 5 line 148 please add space to Figure.3). 4. Whenever possible add 3D structure ID as it has been done for figure 4, otherwise in the corresponding legend to figure indicate reference or alternatively indicate the software used to generate the 3D structure.

Author Response

Response to Reviewer 2’s Comments

Point 1: Figure 1 and that of the Figure 7, please indicate the meaning of the asterisk.

Response 1: Thank you for your comments. We have added the denotation of C* which represents the oxidized Cysteine residue in the figure legends of Figure 1 (page 3 line 78) and Figure 7 (page 12 line 285).

Point 2: It would be appreciable add a scale bar to the Figure 2.

Response 2: A scale bar has been added to Figure 2.

Point 3: A couple of typos need to be edited (e.g., page 5 line 148 please add space to Figure 3).

Response 3: We have reviewed and edited the main text as you instructed.

Point 4: Whenever possible add 3D structure ID as it has been done for figure 4, otherwise in the corresponding legend to figure indicate reference or alternatively indicate the software used to generate the 3D structure.

Response 4: In response to your comment, we have added PDB IDs of UBR box 3D structures in figures or figure legends. In addition, we have added a method section describing how the homology model and the 3D structural graphics were generated.

Reviewer 3 Report

The review by Kim et al. provides an excellent overview of the interaction between UBR box N-recognin and N-degron. The interactions of the N-recognins with their substrates are well described as well their regulatory role in different signaling pathways. The paper is well-written and clear. Particular mention must be made of the number and quality of the figures. in fact, they provide an excellent aid to understanding the complicated mechanisms of interaction between UBR box E3 ligases and their substrates. 

Author Response

Thank you very much for taking the time to review our manuscript.